# Irradiation-induced palladium-catalyzed decarboxylative desaturation enabled by a dual ligand system

Wan-Min Cheng[1], Rui Shang (ID) [1] & Yao Fu[1]

Generation of alkenes through decarboxyolefination of alkane carboxylates has significant synthetic value in view of the easy availability of a variety of carboxylic acids and the synthetic versatility of alkenes. Herein we report that palladium catalysts under irradiation with blue LEDs (440 nm) catalyze decarboxylative desaturation of a variety of aliphatic carboxylates to generate aliphatic alkenes, styrenes, enol ethers, enamides, and peptide enamides under mild conditions. The selection of a dual phosphine ligand system is the key enabler for the successful development of this reaction. The Pd-catalyzed decarboxylative desaturation is utilized to achieve a three-step divergent synthesis of Chondriamide A and Chondriamide C in overall 68% yield from simple starting materials. Mechanistic studies suggest that, distinct from palladium catalysis under thermal condition, irradiation-induced palladium catalysis involves irradiation-induced single-electron transfer and dynamic ligand-dissociation/association process to allow two phosphine ligand to work synergistically.

[1] Hefei National Laboratory for Physical Sciences at the Microscale, CAS Key Laboratory of Urban Pollutant Conversion, Anhui Province Key Laboratory of Biomass Clean Energy, iChEM, Department of Chemistry, University of Science and Technology of China, 230026 Hefei, China. Correspondence and requests for materials should be addressed to R.S. (email: rui@chem.s.u-tokyo.ac.jp) or to Y.F. (email: fuyao@ustc.edu.cn)

**P**roduction of alkenes from carboxylic acids through extrusion of the carboxyl group is a highly significant process[1–5] considering the easy availability of carboxylic acids[6–11] and the synthetic versatility of alkene products[12]. The synthetic significance of this transformation is further enhanced when it is applicable to α-hydroxy acids and α-amino acids to produce enol ethers and enamides, which are important intermediates for synthesis[13,14] and privileged structures in bioactive compounds[15,16]. Reported methods for the transformation of aliphatic carboxylic acids into alkenes include transition-metal-catalyzed decarbonylative dehydration[17–22], decarboxylative oxidation using Pb(IV)[23], and enzymatic processes[24,25]. These methods require either harsh reaction conditions or toxic reagents, or lack generality for synthetic utility. Recently, Glorius and coworkers reported a decarboxylative desaturation reaction of aliphatic redox-active esters by using an organophotoredox catalyst merged with a copper catalyst[26]. With our interest in exploring palladium catalysis under visible-light irradiation[27–32], and also inspired by the elegant achievements of Gevorgyan and coworkers on irradiation-induced palladium-catalyzed desaturation methodologies[33–36], we conceived that a decarboxylative desaturation method should be feasible using irradiation-induced palladium catalysis through hybrid alkyl Pd(I) radical species[37] generated by single-electron transfer (SET)[27–36] activation and radical decarboxylation (Fig. 1). Herein, we report palladium-catalyzed decarboxylative desaturation of various aliphatic carboxylates to generate aliphatic alkenes, styrenes, enol ethers, enamides, and peptide enamides under mild irradiation conditions. Distinct from palladium catalysis under thermal conditions, for which a single phosphine ligand is generally applied[38], a dual phosphine ligand system containing a bidentate phosphine and a bulky monodentate phosphine is the key enabler to achieve this transformation. The reaction demonstrated herein not only provides access to olefins, enol ethers, and enamides from easily available carboxylates, which is of high synthetic value, but also reveals that palladium catalysis under irradiation excitation has distinct ligand requirements compared with traditional thermal systems, and that optimizing a dual ligand combination provides the opportunity to achieve new reactivity.

## Results

**Reaction optimization**. The optimized reaction conditions are summarized in the equation in Table 1. A transparent Schlenk tube charged with palmitic acid-derived redox-active ester (0.2 mmol), PdCl₂ (2 mol%), 4,5-bis(diphenylphosphino)−9,9-dimethylxanthene (Xantphos, 3 mol%), 2-(dicyclohexylphosphino) biphenyl (Cy-JohnPhos, 4 mol%), and 2,4,6-collidine (0.2 mmol) in N,N-dimethylacetamide (DMA) solvent was exposed to irradiation with 30 W blue LEDs at room temperature. After 15 h irradiation and aqueous workup, the desired decarboxylative elimination product **2** was obtained in 93% yield, and only trace amount of decarboxylative protonation by-product **3** was detected. The key results obtained by investigating the controlling parameters of this reaction are summarized in Table 1 (See Supplementary Tables 1–4 for details). Control experiments were conducted to examine the essential role of each parameter of the reaction (entries 1–4). The yield decreased sharply to 44% when Xantphos (3 mol%) was used alone, without Cy-JohnPhos (entry 1). The reaction did not proceed without Xantphos (entry 2). Using a combination of preformed Pd(Xantphos)Cl₂ complex (2 mol%) and Cy-JohnPhos (4 mol%), instead of adding PdCl₂ and each ligand alone, provided similar results. From these results, we can conclude that both Xantphos and Cy-JohnPhos play essential roles in controlling the reactivity of the palladium catalyst. The reaction did not proceed in the absence of irradiation (entry 3). 2,4,6-Collidine was added as base to promote the reaction by reacting with Pd–H formed after β-H elimination to regenerate Pd(0). The efficiency of the palladium catalyst decreased without the addition of 2,4,6-collidine (entry 4). Screening the palladium catalysts showed that PdCl₂ was the optimal catalyst; other palladium catalysts such as Pd(OAc)₂, Pd(TFA)₂, and Pd₂(dba)₃ were ineffective (entries 5−7). Other bases, such as K₂CO₃, DMAP, and Et₃N used instead of 2,4,6-collidine showed poor performance (entries 8−10). When the loading of palladium catalyst was increased to 3 mol%, the necessary amount of the 2,4,6-collidine could be reduced to 25 mol% to give comparable efficiency (entry 11).

The effect of the ligand was investigated, and representative examples are shown in Table 1. Testing Cy-JohnPhos in combination with other bidentate phosphine ligands revealed that the bite angle[39,40] and backbone structure of the bidentate ligand affects the outcome of the reaction significantly. Xantphos was the only effective bidentate ligand, probably because of its large bite angle and conjugated backbone structure. The superiority of Xantphos compared with other bidentate phosphine ligands was also observed in our previously reported alkyl Heck reaction[27] and decarboxylative alkyl Heck reaction[28]. The results of investigating various monodentate phosphine ligands were intriguing. When PPh₃ was used as monodentate ligand instead of Cy-JohnPhos, the reaction was almost completely suppressed. Applying P(1-Np)₃ (1-Np, 1-naphthyl) gave the desired product in 52% yield. Comparing the result using PPh₃ and P(1-Np)₃, it is suggested that the cone angle and the steric bulk of the monodentate phosphine ligands control the reactivity. Guided by this hypothesis and consideration that electronic properties play important role, we further tested other monodentate alkyl phosphine ligands with large cone angles. Bulky monodentate phosphines such as PCy₃ and P(t-Bu)₃ effectively promoted the yields to 60 and 54%, respectively, but using the less bulky ligand PhPCy₂ gave the product in only 12% yield, which is consistent with our hypothesis. Our attention focused on Buchwald phosphine ligands[41], which provide choices to screen ligands of different steric effects and electronic properties. After screening various Buchwald phosphine ligands, it was revealed that too much steric bulk is also detrimental for the reaction (comparing Xphos, Sphos, JohnPhos, and Cy-JohnPhos). Cy-JohnPhos appeared to be an excellent monodentate ligand to promote this reaction, working collaboratively with Xantphos. From the results of the ligand screening study, it is conclusive that both the structures of the ligands and the combination of monodentate and bidentate phosphine ligands determine the efficiency of the palladium catalyst.

**Fig. 1** Working hypothesis. Decarboxylative desaturation via a hybrid alkyl Pd(I) intermediate under irradiation

**Table 1 Parameters affecting the reaction and the effects of the ligand[a]**

**Ligand effect**

a. Bidentate ligand instead of Xantphos

b. Monodentate ligand instead of Cy-Johnphos

| Entry | Variations from optimal conditions | Yield of 2 (%) |
|---|---|---|
| 1 | Without Cy-Johnphos | 44 (6) |
| 2 | Without Xantphos | Trace |
| 3 | Without irradiation | 0 |
| 4 | Without 2,4,6-collidine | 51 |
| 5 | Using Pd(OAc)$_2$ instead of PdCl$_2$ | 10 |
| 6 | Using Pd(TFA)$_2$ instead of PdCl$_2$ | <5 |
| 7 | Using Pd$_2$(dba)$_3$ instead of PdCl$_2$ | <5 |
| 8 | Using K$_2$CO$_3$ instead of 2,4,6-collidine | 30 |
| 9 | Using DMAP instead of 2,4,6-collidine | 43 |
| 10 | Using Et$_3$N instead of 2,4,6-collidine | 40 |
| 11 | Pd(Xantphos)Cl$_2$ (3 mol%), Cy-Johnphos (6 mol%), 2,4,6-collidine (25 mol%) | 92 |

Yields determined by GC analysis with biphenyl as internal standard. Yield shown in parenthesis in entry 1 is the yield of 3
[a]Reaction conditions: aliphatic carboxylate (0.2 mmol), PdCl$_2$ (2 mol%), Xantphos (3 mol%), Cy-JohnPhos (4 mol%), and 2,4,6-collidine (1.0 equiv) in DMA (2 mL), irradiated with 30 W blue LEDs for 15 h under Ar.

**Substrates scope of the decarboxylative desaturation.** The substrate scope of this reaction is demonstrated in Fig. 2. Besides primary aliphatic carboxylates, secondary and tertiary aliphatic carboxylates were also amenable substrates. The mild reaction conditions allowed various functional groups to be tolerated, including ether (**5, 11, 17**), alkyl chloride (**6**), tertiary amine (**6**), sulfonamide (**8**), carboxamide (**9, 10**), ketone (**12, 19**), ester (**18**), and phenolic hydroxyl (**18**). To our delight, oleic acid containing a *Z*-double bond could be efficiently converted into the desired product without *Z*/*E* isomerization (**4**). Notably, allyl arenes were produced without isomerization of the double bond to form thermodynamically favored styryl isomers (**5** and **6**), even in the presence of Pd–H intermediates[42]. α-Aryl aliphatic carboxylates were efficiently converted into the corresponding styrene derivatives in excellent yields, without any byproducts of Heck reactions[28,30–32] or oligomerization[43,44] (**7, 11−14**) being detected. The mild and redox neutral conditions allowed decarboxylative desaturation of mycophenolic acid, containing a phenolic

hydroxyl group, to deliver a 1,3-diene (**18**). In addition, we have demonstrated that natural and pharmaceutical compounds, such as chlorambucil (**6**), gemfibrozil (**17**), and dehydrocholic acid (**19**), were all amenable, providing a useful way to prepare their boronic acid isosteres via hydroboration[45,46]. 1,3-diene and 1,3-enyne could be produced in high yield with *E*-isomer as major isomer (**20, 21**). When an alkyl carboxylate possessing two different eliminable β-H atoms was used, the reaction delivered a mixture of regioisomers with a certain degree of selectivity. High terminal/internal selectivity was achieved for a tertiary cyclic substrate (**9**).

α-Amino acid- and α-hydroxy acid-derived redox-active esters were also amenable substrates (Fig. 3) to generate enol ether and enamide products. In a recent report on photoredox/Cu-catalyzed decarboxylative desaturation, the enol ether product was obtained in low yield (27%) and enamide was not demonstrated[26]. Both tertiary and secondary α-hydroxy carboxylates were suitable substrates, and aryl boronate functionality was well tolerated (**22**,

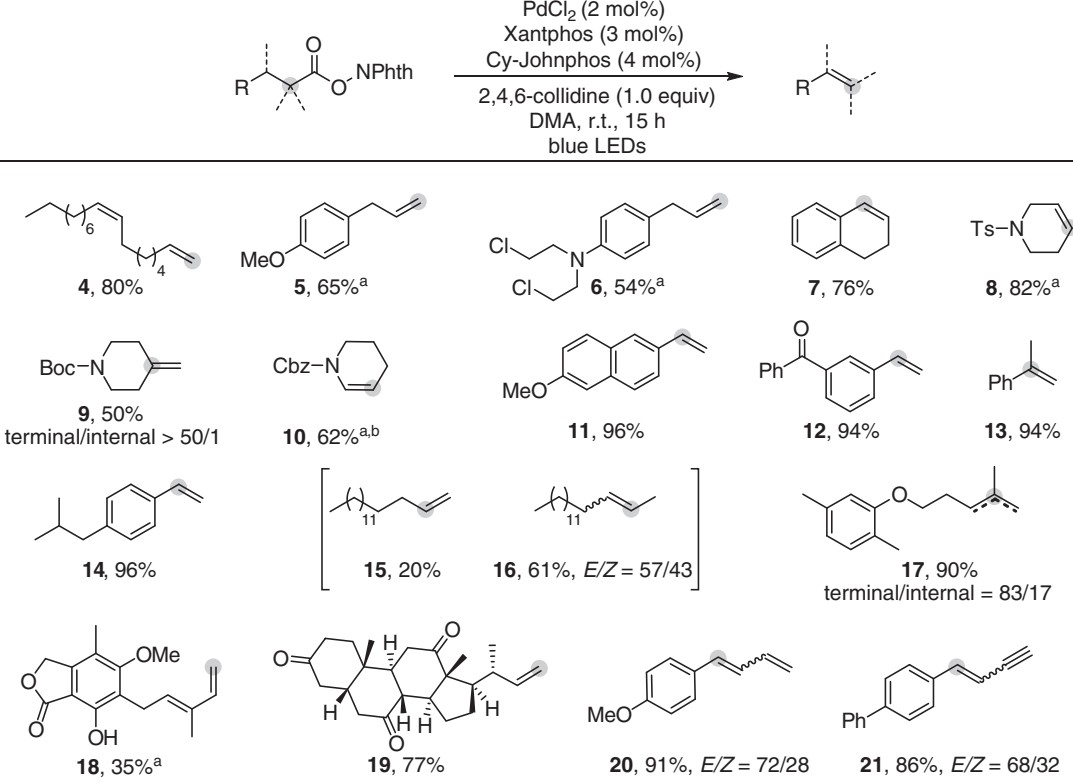

**Fig. 2** Scope of the reaction with respect to aliphatic carboxylates. Reaction conditions: aliphatic carboxylates (0.2 mmol), PdCl$_2$ (2 mol%), Xantphos (3 mol%), Cy-JohnPhos (4 mol%), and 2,4,6-collidine (1.0 equiv) in DMA (2 mL), irradiated with 30 W blue LEDs for 15 h under Ar. Yields of isolated products, the ratio of isomers determined by $^1$H NMR analysis. $^a$PdCl$_2$ (5 mol%), Xantphos (6 mol%), and Cy-JohnPhos (10 mol%) were used. $^b$Benzyl 5,6-dihydropyridine-1(2 H)-carboxylate was obtained in 10%

**23**). 2,3-Dihydro-1-benzofuran-2-carboxylate underwent decarboxylative elimination to yield benzofuran in 82% (**24**). Natural α-amino acid-derived redox-active esters, including proline (**26**), alanine (**27**), phenylalanine (**28**), tryptophan (**29**), methionine (**30**), and glutamic acid (**32**), can be effectively converted into enamides of structural diversity with high *E*-selectivity. Lysine (**33**) derived redox-active ester delivered Boc-protected 2-aminopiperidine because of further intramolecular hydroamination of the enamide product. Protecting groups such as Boc- (**28**), Cbz- (**26**), and Phth- (**31**), used in peptide synthesis, are all compatible. The good performance for various N-protected α-amino carboxylates encouraged us to further test the feasibility of applying this reaction to generate peptide enamides[47]. To our delight, dipeptide- and tripeptide-enamide **34**, **35**, and **36** were successfully synthesized. Considering the diverse transformations of enamide functionality, we believe our method will be useful for peptide modification[48–50].

The synthetic utility of this decarboxylative desaturation is further highlighted by applying it as a key step to simplify the total synthesis of cytotoxic indole-enamide natural products Chondriamides A and C[51,52]. As shown in Fig. 4, Chondriamides A (**37**) and C (**38**) can be prepared at the same time in only three steps from commercially available starting materials in 68% overall yield. The functional group compatibility that enables the free indole N–H bond to be tolerated obviates complex protection/deprotection steps, proving a much simpler and more efficient synthesis of Chondriamides A and C compared with previously established synthesis[47,53].

Interestingly, a ligand-dependent selectivity between β-H elimination and cyclization was observed when testing 2-methyl-5-phenylpentanoic acid-derived redox-active ester (Fig. 5).

The decarboxylative desaturation of this substrate was accompanied by the formation of a large amount of intramolecular C−H radical cyclization product (**39**)[29]. Surprisingly, changing Cy-JohnPhos to a less bulky but more electron-rich trialkyl phosphine, di(1-adamantyl)-*n*-butylphosphine (cataCXium® A), resulted in an overwhelming selectivity for cyclization over elimination. Neither decarboxylative cyclization nor decarboxylative desaturation proceeded in the absence of Xantphos. The ligand-dependent selectivity supports a mechanism involving ligand dissociation because less bulky and electron-rich monodentate ligand may strongly coordinate to palladium in hybrid alkyl Pd(I) radical species, thereby preventing phosphine dissociation/alkyl rebinding to allow β-H elimination to proceed. The other evidence to support a mechanism involving phosphine dissociation to allow β-H elimination is that although the choice of monodentate phosphine significantly affected the yield of elimination product, but it has no obvious effect in selectivity of β-H elimination when tertiary carboxylate possessing distinct β-Hs were tested (Supplementary Table 5). The inefficacy of different monodentate phosphines to affect selectivity of β-H elimination suggests a mechanism involving dissociation of monodentate phosphine before β-H elimination to take place.

## Discussion

UV-Vis spectrum unambiguously confirmed palladium complex is the only light absorption species in the reaction system (Fig. 6a, see Supplementary Figure 80 for details). Stern-Volmer quenching experiment of Pd(PPh$_3$)$_4$ with redox-active ester showed redox-active ester effectively quenches photoexcited Pd(0) species, which supports that a photoexcited Pd(0) species can

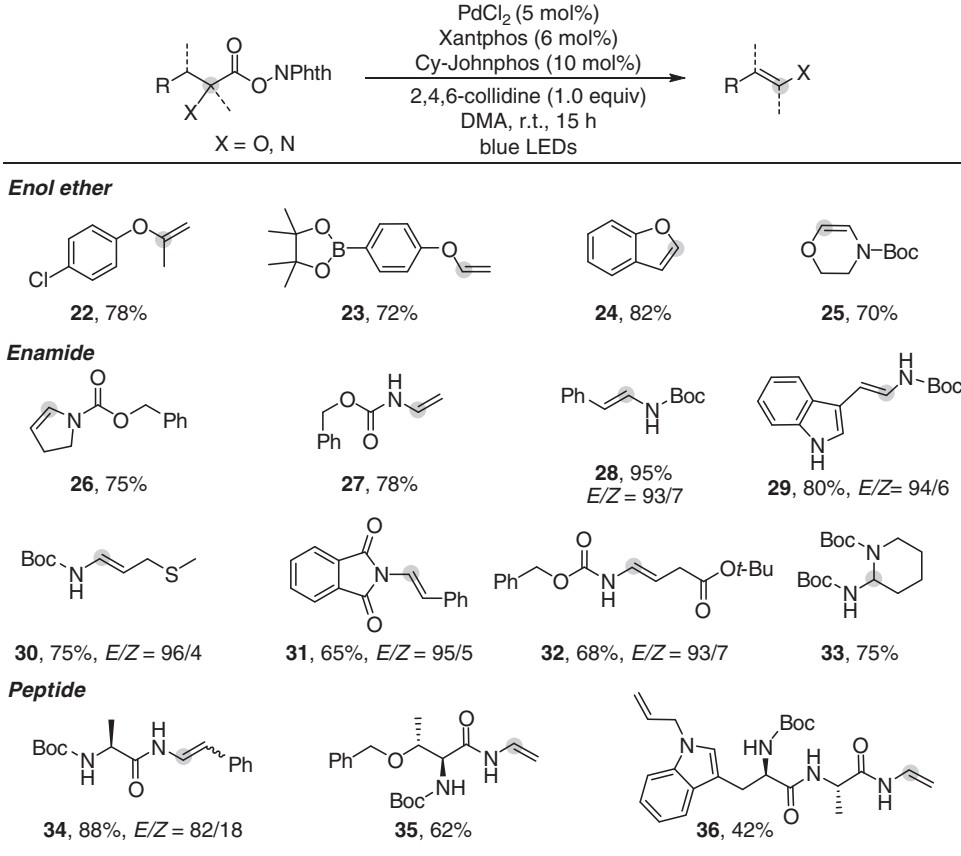

**Fig. 3** Scope of the reaction with respect to α-hydroxy and α-amino carboxylates. Reaction conditions: redox-active esters (0.2 mmol), PdCl₂ (5 mol%), Xantphos (6 mol%), Cy-JohnPhos (10 mol%), and 2,4,6-collidine (1.0 equiv) in DMA (2 mL), irradiated with 30 W blue LEDs for 15 h under Ar. Yields of isolated products, the ratio of isomers determined by ¹H NMR analysis

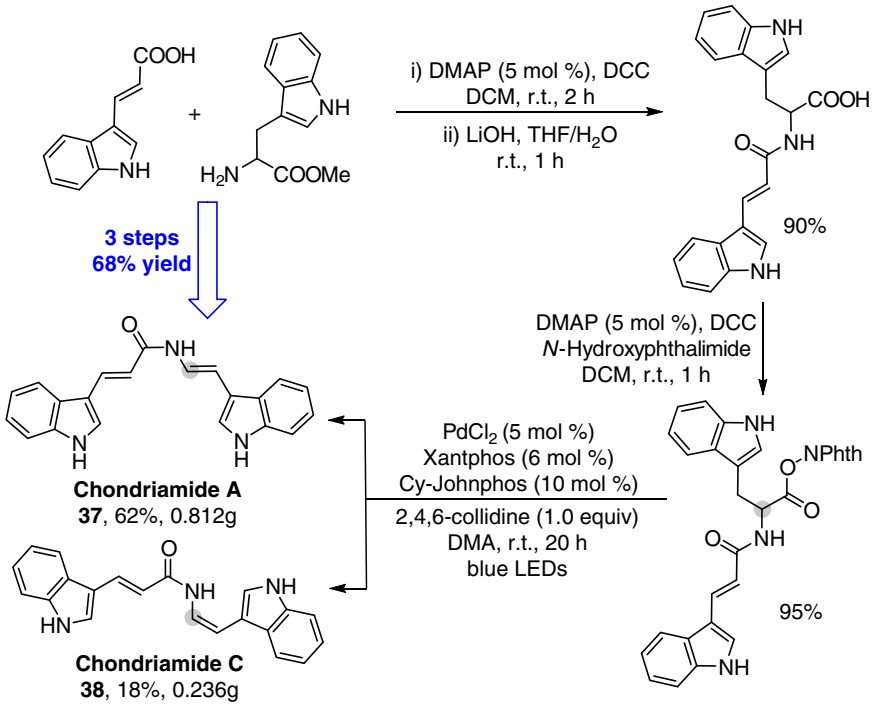

**Fig. 4** Synthesis of Chondriamide A and C. The total synthesis of cytotoxic indole-enamide natural products

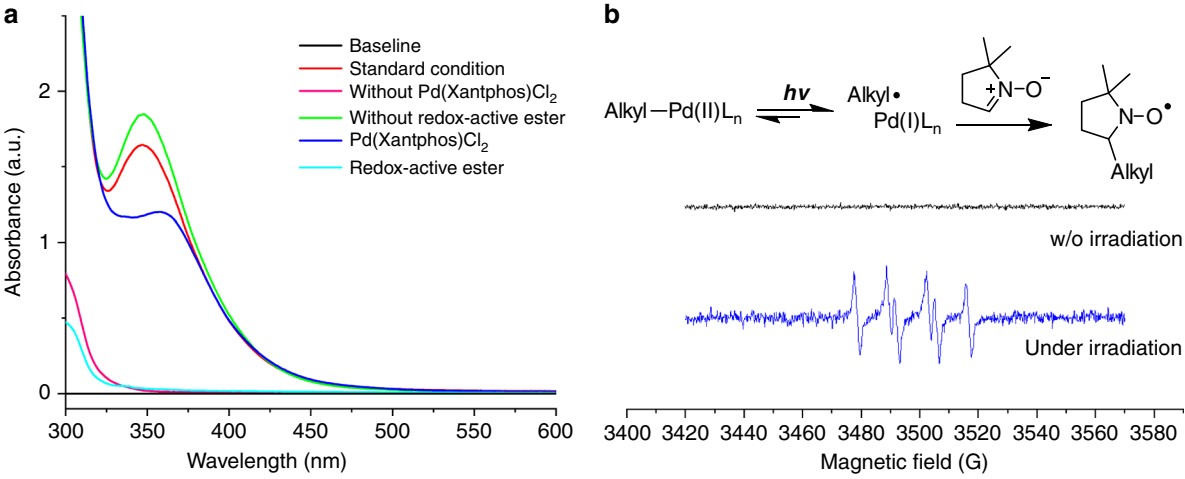

**Fig. 5** Reaction selectivity study. Cyclization/elimination selectivity induced by monodentate phosphine ligands

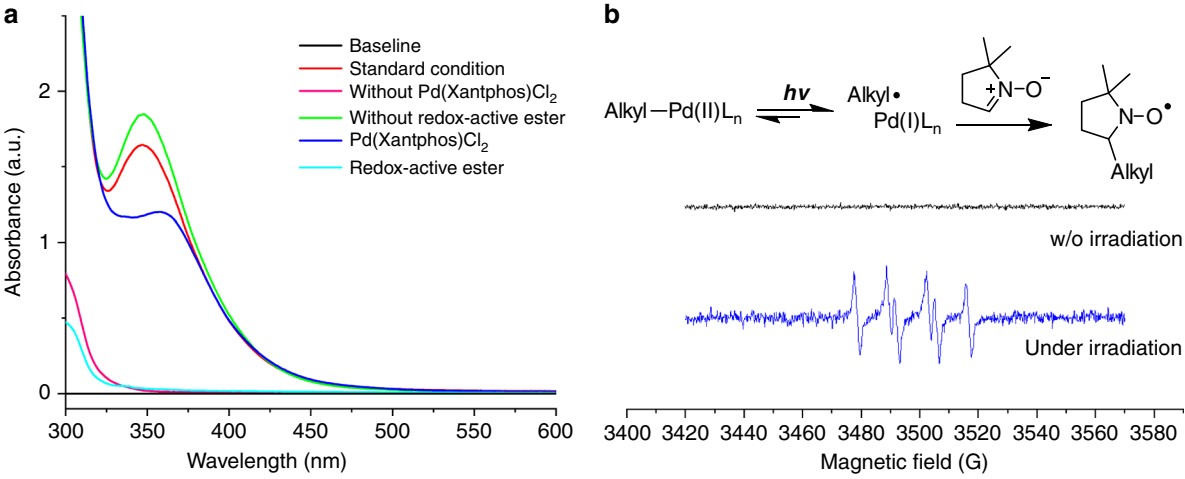

**Fig. 6** Mechanism studies. **a** UV-Vis spectrum of reaction mixture. **b** Electron paramagnetic resonance (EPR) studies of reaction mixtures with and without (w/o) irradiation

transfer electron to activate redox-active ester (See Supplementary Figure 81). However, measurement of Stern-Volmer quenching using a Pd(0)-dual ligand system (Xantphos and Cy-Johnphos) was not successful because of very weak emission generated (~620 nm)[33]. The very weak emission of the Pd(0)-dual ligand system under irradiation compared with Pd(PPh₃)₄ suggests that ligand dissociation or ligand exchange consumes the absorbed energy from blue LED irradiation.

The generation of hybrid alkyl Pd(I) radical species in the reaction system under irradiation was further supported by radical-clock experiments (Fig. 7) and electron paramagnetic resonance (EPR) studies. As depicted in Fig. 6b, EPR signals with hyperfine coupling constant indicating the formation of a spin adduct of alkyl radical with 5,5-dimethyl-1-pyrroline N-oxide (DMPO) was observed[29] (See Supplementary Figure 82 for details). This signal was only observable after irradiation with blue LEDs, which is consistent with our understanding that activation of the redox-active ester and formation of hybrid alkyl Pd(I) species requires irradiation[28]. Inspired by these results and recent reports by Gevorgyan et al.[30,31,33–36]., we may conclude Pd-alkyl bond under blue LED irradiation exhibits radical property, which resembles to reported radical properties of alkyl-Ni and alkyl-Fe species under thermal conditions[54]. EPR measurements were further used to obtain mechanistic understandings of the dual ligand effect in this transformation. EPR signals of spin adduct formed by trapping alkyl radical with DMPO were also observed when either Pd (Xantphos)Cl₂ alone or PdCl₂/Cy-JohnPhos was used as catalyst. The EPR results suggest that the dual ligand system is not essential for SET-activation of redox-active ester under irradiation, but may facilitate it (refer to Table 1, entries 1 and 2). The important role of the dual ligand may be related with stabilization of Pd(I) intermediate (e.g. **B** and **C** in Fig. 8) before alkyl binding to enable desired catalytic cycle in Fig. 8.

Our mechanistic insights regarding to the irradiation effect and the unusual dual ligand effect on palladium catalysis are provided in Fig. 8. Based on our mechanistic studies and our previous reports[27–29], we considered irradiation induced palladium catalysis involves irradiation-induced single-electron transfer of a dual phosphine-coordinated Pd(0) complex (**A**) with substrate, and irradiation-induced ligand dissociation/association pathways. As depicted in Fig. 8, in this reaction, we hypothesized that a palladium (0) complex coordinated with both Xantphos and Cy-JohnPhos transfers electron to redox-active ester to induce decarboxylation to form a hybrid alkyl Pd(I) intermediate. The alkyl Pd(I) intermediate possessing two types of phosphine ligand can dissociate one weakly coordinate phosphine ligand under irradiation to allow alkyl binding to undergo β-H elimination. The dissociated monodentate phosphine rebinds to Pd(0) catalyst after releasing olefin product. Accordingly to our observation, while the bidentate phosphine (Xantphos) may be related with photoexcitation, appropriate choice of monodentate phosphine of suitable association and dissociation ability is the key to tune the reactivity of hybrid alkyl Pd(I) intermediate to enable new catalytic transformations.

In summary, we report herein palladium-catalyzed decarboxylative desaturation reaction enabled by a dual ligand system under mild irradiation conditions. The reaction provides a useful method to access various alkenes, including enol ethers, enamides, and peptide enamides, from easily available carboxylates. Synthesis of Chondriamide A and Chondriamide C can be simplified by applying this method. The unusual ligand effect observed herein reveals two phosphine ligands work synergistically through an association/dissociation process under irradiation, and points out opportunity to discover untapped reactivity of irradiation-induced palladium catalysis by exploring dual ligand combinations.

**Fig. 7** Radical-clock experiments. Ring-opening/ring-closure experiments supporting the radical property of alkyl-palladium species under irradiation

**Fig. 8** Proposed mechanism. Mechanistic insights regarding to irradiation effect and dual ligand effect

## Methods

**General procedure for the decarboxylative desaturation**. Redox-active ester (1.0 equiv, 0.2 mmol) (if solid), PdCl$_2$ (2 mol%, 0.7 mg), Xantphos (3 mol%, 3.5 mg), and Cy-Johnphos (4 mol%, 2.8 mg) were placed in a transparent Schlenk tube equipped with a stirring bar. The tube was evacuated and filled with argon (three times). To these solids, redox-active ester (1.0 equiv, 0.2 mmol) (if liquid), anhydrous N,N-dimethylacetamide (DMA, 2.0 mL) and 2,4,6-collidine (1.0 equiv, 0.2 mmol, 24.2 mg) was added via a gastight syringe under argon atmosphere. The reaction mixture was stirred under the irradiation of Kessil blue LEDs (distance app. 3.0 cm from the bulb) at room temperature for 15 h. After 15 h, the mixture was quenched with saturated NaCl solution and extracted with ethyl acetate for three times. The organic layers were combined and concentrated under vacuo. The product was purified by flash column chromatography on silica gel with petroleum ether or a mixture of petroleum ether and ethyl acetate as eluent. The NMR spectra of all products can be found in Supplementary Figures 1–79.

## Data availability

The authors declare that all the data supporting the findings of this study are available within the paper and its supplementary information files, or from the corresponding author upon request.

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

## Acknowledgements

This work was supported by National Key R&D Program of China (2017YFA0303502), National Natural Science Foundation of China (21325208 and 21572212), Strategic Priority Research Program of CAS (XDB20000000 and XDA21060101), Major Program of Development Foundation of Hefei Center for Physical Science and Technology (2017FXZY001), and KY (2060000119). W.-M.C. thanks China Postdoctoral Science Foundation (2017M622004) and Anhui Provincial Natural Science Foundation (1808085QB43).

## Author contributions

R.S. conceived the idea and wrote the manuscript. W.M.C. performed the experiments. W.M.C. and Y.F. analyzed the data and participated in the preparation of the manuscript.

## Additional information

**Competing interests:** The authors declare no competing interests.

