## [Peer Review File · Nature Communications]

Reviewers' comments:

Reviewer #1 (Remarks to the Author):

Shang, Fu and coworkers reported a Palladium-Catalyzed Decarboxylative Desaturation of aliphatic carboxylates to generate alkenes. The author also shown the application of this method to the synthesis of natural products and conduct some experiments towards possible radical species. However, as author mentioned, Glorius and coworkers reported a similar result by using an organophotoredox catalyst merged with a copper catalyst (ref. 26). And also, the irradiation-induced palladium-catalyzed

desaturation methodologies have been well developed by other groups, such as Prof. Gevorgyan (ref. 33-35). Although the author carried out the experiments to confirm the Pd(I) radical species, this process has been reported by previously reported works. (Prof. Gevorgyan and Prof. Yu (Angew. Chem. Int. Ed. 2017, 56, 15683.)). A dual ligand system can assist the decarboxylative desaturation, the exact role can be clarified is better for this research. How about the synthesis of multi-substituted alkenes? If the ligands can control the selectivities on preparing tetra-substituted alkenes, the utility and the novelty of this research will be very well. Based on these results, this reviewer think that the novelty of this work is not suitable for publish on Nat. Commun.

Other questions:

1. From line 98 to line 106, The author comment the effect of monodentate phosphine ligand on the cone angle, may be the electronic properties are also play an important role for this transformation. (looks like electron rich monodentate phosphines are better than the electron poor ones).

Line 105 "PdPCy2" changed to "PhPCy2"

2. If a tertiary aliphatic carboxylate with primary, secondary, and tertiary C-H bonds at β -position, how about the selectivity?

3. Line 145, Is the structure of 31 correct?

Reviewer #2 (Remarks to the Author):

The authors report a Pd-catalyzed photoinduced decarboxylative olefination of activated carboxylic ester derivatives with a dual ligand system.

There are two aspects to this manuscript: 1) the synthetic utility of the transformation itself, and 2) the intriguing mechanistic implications.

I can not recommend the manuscript for publication in Nat Chem for synthetic utility alone due to the published work that is very similar, for example the Glorius work, as appropriately referenced in this manuscript.

However, the combination of a Pd catalyst and photoredox chemistry, and the intriguing requirement for the dual catalyst system is of fundamental interest that may warrant publication in a higher impact factor journal, in my opinion. The implication of these results to the larger field may be of high interest. As of now, the results pertaining to mechanism are more treated as an afterthought to a synthetic paper. That in my opinion is not sufficient, and not the right focus.

The epr study is a good start, but not much more. Observing an epr signal in the presence of a radical trap and irradiation is no surprise; not observing a signal would be. There is no quantification, interpretation or valuable analysis.

Also, the requirement for two different phosphine ligands is highly unusual. The simple hypothesis that "one ligand must dissociate" is neither satisfying nor justified based on the data provided.

I could describe a large list of appropriate mechanism experiments, but it is, in my opinion, really up to the authors to better make a case of what is the relevant information in this transformation.

With a few more experiments and experimentally verified claims regarding the role of the different ingredients, I think the manuscript would make an interesting addition to Nat Chem.

Reviewer(s)' Comments to Author:

Reviewer: 1

Shang, Fu and coworkers reported a Palladium-Catalyzed Decarboxylative Desaturation of aliphatic carboxylates to generate alkenes. The author also shown the application of this method to the synthesis of natural products and conduct some experiments towards possible radical species. However, as author mentioned, Glorius and coworkers reported a similar result by using an organophotoredox catalyst merged with a copper catalyst (ref. 26).

Response: We thank this reviewer to critically compare our work with related literatures. The beautiful work by Glorius and co-workers indeed provided similar synthetic transformation to synthesis alkenes from carboxylates, while a major part in our work focused on synthesizing enamides and peptide enamides, which

were not achieved previously. We added in this revision that our method can also be used to produce 1,3-diene and 1,3-enyne in high yields.

From a viewpoint of catalysis, our method provides entirely different mechanistic scenario compared with Glorius's organophotoredox/copper relay catalysis.

And also, the irradiation-induced palladium-catalyzed desaturation methodologies have been well developed by other groups, such as Prof. Gevorgyan (ref. 33-35). Although the author carried out the experiments to confirm the Pd(I) radical species, this process has been reported by previously reported works. (Prof. Gevorgyan and Prof. Yu (Angew. Chem. Int. Ed. 2017, 56, 15683.). A dual ligand system can assist the decarboxylative desaturation, the exact role can be clarified is better for this research.

Response: Irradiation-induced palladium-catalyzed desaturation methods have been elegantly developed by Gevorgyan and co-workers, but these methods are not decarboxylative desaturation. The mechanistic understanding of irradiation-induced palladium catalysis is still in its infancy. Our studies is in accordance with reports by Gevorgyan *et al.* considering Pd(I) species as hybrid aryl/alkyl Pd(I) radical species, while our investigation is hardly in agreement with the mechanistic conclusion drawn by Yu, D. G. and co-workers, which claimed palladium works as a photoredox catalyst. In our report, we first pointed out the Pd (I) species generated under irradiation has its distinct ligand preference compared with ligand effect observed in traditional Pd(0)/Pd(II) catalysis, that can enable new catalytic reactivity. We further clarified this point in our revised manuscript.

How about the synthesis of multi-substituted alkenes? If the ligands can control the selectivities on preparing tetra-substituted alkenes, the utility and the novelty of this research will be very well. Based on these results, this reviewer think that the novelty of this work is not suitable for publish on Nat. Commun.

Response: We thank this reviewer for this suggestion. We demonstrated synthesis of tri-substituted alkene with terminal selectivity in Scheme 1, entries 9 and 17.

We tested 2,3-dimethyl-2-phenylbutanoate, but this substrate couldnot deliver tetra-substituted olefin due to large steric hindrance at alfa-position.

We are afraid to say the synthesis of tetra-substituted olefin using our method requires very special tertiary carboxylate with branched alkyl substituent at alfa-position, which may limit the synthetic value.

Other questions:

1. From line 98 to line 106, The author comment the effect of monodentate phosphine ligand on the cone angle, may be the electronic properties are also play an important role for this transformation. (looks like electron rich monodentate phosphines are better than the electron poor ones).

Response: By comparing the results using PPh₃ and P(Np)₃, which has similar electronic properties but different cone angle, we can conclude the sterics of monodentate ligands affect reaction outcomes. Most of the electron rich alkyl phosphines have large cone angles. We agree with this reviewer the electronic properties of monodentate ligand also play important role. We addressed this comment in our revised manuscript.

Line 105 "PdPCy2" changed to "PhPCy2"

Response: We corrected this typo.

2. If a tertiary aliphatic carboxylate with primary, secondary, and tertiary C-H bonds at β-position, how about the selectivity?

Response: We tested this suggestion using 2,3-dimethyl-2-phenylbutanoate, but this substrate didnot deliver tetra-substituted olefin due to large sterics at alfa-position. For secondary aliphatic carboxylate, we observed clearly terminal selectivity, which supported mechanism of beta-H elimination rather than E1 elimination via carbenium cation.

3. Line 145, Is the structure of 31 correct?

Response: It is correct.

Reviewer: 2

The authors report a Pd-catalyzed photoinduced decarboxylative olefination of activated carboxylic ester derivatives with a dual ligand system.

There are two aspects to this manuscript: 1) the synthetic utility of the transformation itself, and 2) the intriguing mechanistic implications.

I can not recommend the manuscript for publication in Nat Chem for synthetic utility alone due to the published work that is very similar, for example the Glorius work, as appropriately referenced in this manuscript.

Response: We are very much afraid that one major synthetic advantage of our chemistry might got neglected, that is the reaction can be applied to the synthesis of enamides and peptide enamides, which are valuable synthetic intermediates, but has not been achieved in Glorius's work using organophotoredox/copper catalyst. Because of the capability to make enamide in high yield, we were able to apply our method in the short-step total synthesis of natural products, Chondriamide A and C in high yield. In addition, we add two examples of synthesizing conjugated 1,3-diene and 1,3-enyne through internal decarboxylative elimination in our revised manuscript.

However, the combination of a Pd catalyst and photoredox chemistry, and the intriguing requirement for the dual catalyst system is of fundamental interest that may warrant publication in a higher impact factor journal, in my opinion. The implication of these results to the larger field may be of high interest. As of now, the results pertaining to mechanism are more treated as an afterthought to a synthetic paper. That in my opinion is not sufficient, and not the right focus.

The epr study is a good start, but not much more. Observing an epr signal in the presence of a radical trap and irradiation is no surprise; not observing a signal would be. There is no quantification, interpretation or valuable analysis.

Also, the requirement for two different phosphine ligands is highly unusual. The simple hypothesis that "one ligand must dissociate" is neither satisfying nor justified based on the data provided.

I could describe a large list of appropriate mechanism experiments, but it it, in my opinion, really up to the authors to better make a case of what is the relevant information in this transformation.

With a few more experiments and experimentally verified claims regarding the role of the different ingredients, I think the manuscript would make an interesting addition to Nat Chem.

Response: We appreciate these excellent comments by this reviewer. Based on these insightful comments, we furthered conducted EPR experiments to study the ligand effect on single-electron-transfer activation of redox active ester, as well as the dual ligand effect on the selectivity of beta-H elimination. We provided our mechanistic understanding in the revised manuscript, claiming "Mechanistic studies suggest that, distinct from palladium catalysis under thermal condition, irradiation-induced palladium catalysis involves irradiation-induced single-electron transfer and dynamic ligand-dissociation/association process to allow two phosphine ligand to work synergistically."

A mechanistic scheme is added to the revised manuscript to clearly illustrate our understanding of the dual ligand effect on the catalytic cycle.

We hope the revised manuscript incorporated with all the necessary changes and additions will be suitable for publication.

Reviewers' comments:

Reviewer #2 (Remarks to the Author):

I encouraged the execution of meaningful additional experiments in my previous evaluation. Unfortunately, the authors did not execute these experiments. What has been done is a few not very meaningful additions that are not on par with state of the art analyses. What is absorbing the photons, the Pd or the redox-active ester? Where is the Stern Volmer plot? Is the epr study quantified? The authors write: "The generation of hybrid alkyl Pd(I) radical species in the reaction system under irradiation was further supported by electron paramagnetic resonance (EPR) studies". Do the authors even understand what they are writing? What is a "hybrid alkyl Pd(I) radical species". There is no such thing (or at least nobody has ever observed anything similar and published it in the literature as of yet, to the best of my knowledge. Every Pd d9 would immediately recombine with a C-based radical, so observing an eps for the intermediate is extremely unlikely.

I apologize to write such a negative review, but the authors seem to have no understanding of what they are trying to study or how one would go about it. It remains a synthetic paper, without meaningful mechanism experiments. That alone, in my opinion is not sufficient for publication in this journal. I had expressed substantial enthusiasm because I think the results are potentially very interesting and relevant, but the authors were not able to address the comments I provided with respect to mechanism. What is provided now with respect to mechanism analysis is fundamentally flawed and confusing and should not be published.

Reviewers' comments:

Reviewer #2 (Remarks to the Author):

I encouraged the execution of meaningful additional experiments in my previous evaluation. Unfortunately, the authors did not execute these experiments. What has been done is a few not very meaningful additions that are not on par with state of the art analyses.

What is absorbing the photons, the Pd or the redox-active ester?

Response: We demonstrated in our previous work (and it is well reported) that redox-active ester does not have absorption under visible light (455 nm). Please notice we mentioned this in our manuscript and cited the precedented reports (ref. 28). It is clear that Pd complex absorbs photons. In the revision, we added UV-Vis spectrum to clearly show that Pd complex is the light absorbing species.

Where is the Stern Volmer plot?

Response: It is probably necessary to point out that palladium complex to be photo-activated is not similar with dyes (or photoredox catalysts which have rigid structure), which have emission to be measured by Stern-Volmer quenching experiments. The irradiation in our chemistry is exerting a scission effect to weaken palladium-carbon bond. The irradiation causes a scission effect to cleave chemical bond, so *under the dual ligand system*, we did not observed obvious fluorescence or phosphorescence emission to allow measurement of Stern-Volmer plot. We added Stern-Volmer quenching experiments using Pd(PPh₃)₄ and redox-active ester in the SI. The results support that photoexcited Pd(0) species can be quenched by redox active ester.

Is the epr study quantified? The authors write: "The generation of hybrid alkyl Pd(I) radical species in the reaction system under irradiation was further supported by electron paramagnetic resonance (EPR) studies".

Do the authors even understand what they are writing? What is a "hybrid alkyl Pd(I) radical species". There is no such thing (or at least nobody has ever observed anything similar and published it in the literature as of yet, to the best of my knowledge. Every Pd d9 would immediately recombine with a C-based radical, so observing an eps for the intermediate is extremely unlikely.

Response: We respectfully disagree with the reviewer's comment on "hybrid alkyl Pd(I) radical species". Hybrid aryl, vinyl, or alkyl Pd(I) radical species have, at least, been published several times in JACS and Angew by Gevorgyan and co-workers recently. It is a species generated by

photo-scission of covalent Pd(II)-carbon bond. The literatures reporting these species are: *J. Am. Chem. Soc.* **2016**, *138*, 6340.; *J. Am. Chem. Soc.* **2017**, *139*, 14857.; *Angew. Chem. Int. Ed.* **2018**, *57*, 2712.; and *J. Am. Chem. Soc.* **2018**, *140*, 2465.

The referee comments "Every Pd d9 would immediately recombine with a C-based radical, so observing an eps for the intermediate is extremely unlikely." is probably correct for reactions under thermal conditions, but for reactions under irradiation, not every Pd d9 would immediately recombine with carbon-based radicals as clearly observed in Gevorgyan's reports and our early report last year (*J. Am. Chem. Soc.* **2017**, *139*, 50, 18307). Please notice we used DMPO (5,5-dimethyl-1-pyrroline N-oxide) to react with hybrid alkyl Pd(I) radical and the EPR signal of spin adduct formed by trapping alkyl radical with DMPO was clearly observed. Although we did not quantify the EPR measurement (we do not know how to quantify it in this DMPO spin trapping experiment), the EPR signal is in accordance with the signal of spin adduct formed by trapping radical with DMPO reported in literatures (*J. Org. Chem.* **2005**, *70*, 1198). *Reviewer 2 might misunderstand that the EPR signal is from alky palladium species.* The EPR signal shown in Figure 2b is from alkylated DMPO radical. We added the equation of DMPO reacting with alkyl palladium species under irradiation in Figure 2b to avoid misunderstanding.

In addition, we added radical clock experiments to further confirm the radical property of alkyl palladium species under irradiation of blue LEDs.

We thank the insightful comments reviewers 2 provided in the previous review and we hope reviewer 2 could reevaluate the above reviewer comments and our explanation.

I apologize to write such a negative review, but the authors seem to have no understanding of what they are trying to study or how one would go about it. It remains a synthetic paper, without meaningful mechanism experiments. That alone, in my opinion is not sufficient for publication in this journal. I had expressed substantial enthusiasm because I think the results are potentially very interesting and relevant, but the authors were not able to address the comments I provided with respect to mechanism. What is provided now with respect to mechanism analysis is fundamentally flawed and confusing and should not be published.

Response: We thank this reviewer for his/her harsh criticism to make us aware the necessity to further clarify our results in revised manuscript. We sincerely hope this reviewer could reevaluate our results and his/her comments.

Thank you for your consideration, and we look forward to hearing from you soon.

With best regards,

Dr. Rui Shang

The University of Tokyo

Hongo, Bunkyo-ku, Tokyo 113-0033, Japan

E-mail: ruishang@chem.s.u-tokyo.ac.jp

REVIEWERS' COMMENTS:

Reviewer #2 (Remarks to the Author):

The authors have written a strong response to my critique. In fact, the response letter, in my opinion, is much clearer than the manuscript is (was). The authors have added several sentences that clarify what is observed and how it is substantiated. I have to say that I did in fact misunderstand the original claims in the context of what was reported in the manuscript. I now understand, and I agree with the comments. I would suggest that the authors clarify exactly in the manuscript what can be claimed and what is a hypothesis to avoid confusion. The authors have added experimental detail, and I no longer complain about the actual experiments (or lack thereof).

I recommend acceptance of the manuscript.

REVIEWERS' COMMENTS:

Reviewer #2 (Remarks to the Author):

The authors have written a strong response to my critique. In fact, the response letter, in my opinion, is much clearer than the manuscript is (was). The authors have added several sentences that clarify what is observed and how it is substantiated. I have to say that I did in fact misunderstand the original claims in the context of what was reported in the manuscript. I now understand, and I agree with the comments. I would suggest that the authors clarify exactly in the manuscript what can be claimed and what is a hypothesis to avoid confusion. The authors have added experimental detail, and I no longer complain about the actual experiments (or lack thereof). I recommend acceptance of the manuscript.

Response: We appreciate the reviewer's efforts to further evaluate our revised manuscript and response letter. According to the reviewer's suggestion, we have modified the relevant sentence as "we hypothesized that a palladium (0) complex coordinated with both Xantphos and Cy-JohnPhos transfers electron to redox-active ester to induce decarboxylation to form a hybrid alkyl Pd(I) intermediate".